# Pinocembrin Downregulates Vascular Smooth Muscle Cells Proliferation and Migration Leading to Attenuate Neointima Formation in Balloon-Injured Rats

**DOI:** 10.3390/biom15091325

**Published:** 2025-09-17

**Authors:** Hyeonhwa Kim, Jihye Jung, Young-Bob Yu, Dong-Hyun Choi, Leejin Lim, Heesang Song

**Affiliations:** 1Department of Biomedical Sciences, Chosun University Graduate School, Gwangju 61452, Republic of Korea; sujin1720@chosun.ac.kr (H.K.); skydbs113@chosun.ac.kr (J.J.); 2Department of Paramedicine, Nambu University, Gwangju 62271, Republic of Korea; ybyu@nambu.ac.kr; 3Department of Internal Medicine, Chosun University School of Medicine, Gwangju 61452, Republic of Korea; dhchoi@chosun.ac.kr; 4Advanced Cancer Controlling Research Center, Chosun University, Gwangju 61452, Republic of Korea; 5Department of Biochemistry and Molecular Biology, Chosun University School of Medicine, Gwangju 61452, Republic of Korea

**Keywords:** migration, proliferation, neointima formation, vascular smooth muscle cells, pinocembrin

## Abstract

The abnormal proliferation and migration of vascular smooth muscle cells (VSMCs) are a primary cause of cardiovascular diseases such as atherosclerosis and restenosis after angioplasty. Pinocembrin (5,7-dihydroxyflavanone, PCB), a natural flavonoid compound found abundantly in propolis, has been reported to have antibacterial, anti-inflammatory, antioxidant, and anticancer effects, and cardiac function improvement in ischemic heart disease. In this study, the protective effects of PCB against the migration and proliferation of VSMCs were investigated. MTT and BrdU assays were performed to estimate the cytotoxicity and cell proliferative activity of PCB, respectively. Rat aortic VSMC migrations and neointima formation were evaluated using wound healing, boyden chamber assays, and in balloon-injured (BI) rat, respectively. PCB suppressed the phosphorylated levels of p38 in PDGF-BB-induced VSMCs followed by reducing the expression of MMP2 and 9. PCB downregulated the expression levels of cell cycle regulatory proteins such as PCNA, CDK2, CDK4, and Cyclin D1. Furthermore, the phosphorylated levels of FAK at Y397 and Y925 sites and the expression levels of FAK-related proteins such as Integrin β1, Paxillin, Talin, and Vinculin were significantly reduced by PCB in PDGF-BB-induced VSMCs. The neointima formation was markedly decreased by PCB administration in the carotid artery of a balloon-injured rat. In conclusion, PCB inhibits the proliferation and migration of VSMCs by stimulation of PDGF-BB through the regulation of the p38 and FAK signaling pathway. Therefore, PCB may be a promising therapeutic candidate for preventing and treating cardiovascular diseases such as atherosclerosis and restenosis.

## 1. Introduction

Atherosclerosis is a chronic and progressive inflammatory vascular disease characterized by the accumulation of lipids, fibrous elements, and immune cells within the arterial wall. It begins with endothelial injury induced by various hazardous stimuli such as hyperlipidemia, smoking, oxidative stress, or hypertension [1]. Damaged endothelial cells lose their barrier function and initiate an inflammatory cascade, attracting monocytes and platelets to the lesion site. These cells release pro-inflammatory cytokines and growth factors, further exacerbating vascular inflammation and contributing to plaque formation [2,3].

Vascular smooth muscle cells (VSMCs) play a crucial role in the progression of atherosclerosis. Under pathological conditions, VSMCs migrate from the media to the intima, where they undergo phenotypic switching from a contractile to a synthetic state, characterized by increased proliferation and extracellular matrix (ECM) remodeling [4,5]. This process is mediated, in part, by matrix metalloproteinases (MMPs), especially MMP2 and MMP9, which degrade ECM components and facilitate VSMC migration [6,7,8]. In particular, the platelet-derived growth factor (PDGF), secreted by activated platelets and endothelial cells, enhances MMP expression in VSMCs and promotes their proliferation and motility [4,5,6,7,8,9]. Given the central role of VSMCs in atherosclerotic plaque development, therapeutic strategies targeting their migration and proliferation have gained attention. Although several synthetic drugs have been developed to inhibit these processes, many are associated with adverse effects, including hepatic toxicity, metabolic disturbances, and gastrointestinal issues, which limit long-term use. Consequently, there is growing interest in identifying natural compounds with therapeutic potential that are both effective and biocompatible, offering a safer alternative for atherosclerosis treatment [10,11].

Pinocembrin (5,7-dihydroxyflavanone, PCB) is a major flavonoid monomer found in propolis, a natural substance produced by honeybees, and is also present in various plants such as ginger, fruits, nuts, seeds, herbs, and spices [12,13]. PCB exhibits a wide range of pharmacological properties, including antibacterial, antiviral, anti-inflammatory, antifibrotic, antioxidant, and anticancer activities. It also plays a neuroprotective role, particularly in cerebral ischemia [13,14,15,16,17,18,19,20]. In recognition of its therapeutic potential, PCB has been approved as a novel drug for ischemic stroke by the Chinese Food and Drug Administration and is currently undergoing phase II clinical trials [21].

Recent studies have demonstrated the cardioprotective effects of PCB, including improved cardiac function, reduced infarct size, and suppression of ventricular arrhythmias. These effects are believed to occur through modulation of inflammatory pathways such as NF-κB/TNF-α and reduction in autonomic dysfunction, ultimately attenuating susceptibility to atrial fibrillation [22,23,24]. Additionally, PCB has been shown to relax vascular smooth muscle in rat aortic rings, lower blood lipid levels, and preserve endothelial cell integrity [25]. Combined administration of PCB and simvastatin in apoE-/- mice on a high-fat diet reduced serum lipids, protected against cholesterol-induced endothelial damage, and synergistically inhibited atherosclerotic plaque formation [26].

In cancer studies, PCB has been reported to inhibit cell proliferation, migration, and promote apoptosis through pathways such as FAK/p38 and PI3K/AKT, with notable effects on MMP2/9 expression [27,28]. Despite these well-documented bioactivities, the effect of PCB on vascular smooth muscle cell (VSMC) behavior—particularly their proliferation and migration, key processes in atherosclerosis—remains unexplored. Therefore, this study aims to investigate the inhibitory effects and underlying mechanisms of PCB on PDGF-BB-induced VSMC proliferation and migration, as well as its potential to suppress neointimal hyperplasia in a rat carotid artery balloon-injury model.

## 2. Materials and Methods

### 2.1. Materials

PCB (C_15_H_12_O_4_; molecular weight: 256.25; purity: high-performance liquid chromatography ≥98%) was obtained from Sigma-Aldrich (St. Louis, MO, USA), and its chemical structure is shown in Figure 1A.

### 2.2. Animals

All animal experiments including the isolation of rat aortic VSMCs and balloon-injured carotid arteries rat model were conducted in accordance with the International Guide for the Care and Use of Laboratory Animals. The protocol was approved by the Animal Research Committee of the Chosun University School of Medicine (Protocol No. CIACUC2020-S0032).

### 2.3. Isolation of VSMCs

Primary vascular smooth muscle cells (VSMCs) were obtained from the thoracic aortas of 6-week-old Sprague–Dawley rats, following a modified protocol based on a previously reported method [29]. In brief, excised aortas were carefully cleaned of connective tissue and residual blood, then cut into smaller segments and placed in a digestion solution containing collagenase type I (1 mg/mL; Sigma, St. Louis, MO, USA) and elastase (0.5 mg/mL; Worthington, NJ, USA) for 30 min at 37 °C. The adventitial layer was gently removed under a dissecting microscope, and the aortic segments were incubated again in 5 mL of the same enzyme mixture for an additional 2 h at 37 °C. Tissue dissociation was aided by gentle pipetting, after which the cell suspension was centrifuged at 1600× *g* for 5 min. Pelleted cells were resuspended in Dulbecco’s modified Eagle medium (DMEM; WelGENE, Seoul, Republic of Korea) supplemented with 10% fetal bovine serum (FBS), and the digestion process was repeated until no tissue fragments remained. Cultures were maintained at 37 °C in a humidified incubator (95% air, 5% CO_2_), and cells from passages 1–10 were used for experiments.

### 2.4. Balloon Injury Model

Male Sprague–Dawley rats (6–7 weeks old, body weight 200 ± 50 g) were anesthetized via intramuscular injection of Zoletil 50^®^ (20 mg/kg; Virbac Corp., Fort Worth, TX, USA) and Rompun^®^ (10 mg/kg; Bayer Corp., Pittsburgh, PA, USA). Following a midline neck incision, the left external carotid artery was exposed and injured using a 2-F balloon catheter (Edwards Lifesciences Corp., Irvine, CA, USA) inserted through the aortic outlet of the common carotid artery. The balloon was inflated and deflated to expand the artery and denude the endothelium, and this procedure was repeated three to four times. The catheter was then removed, the external carotid artery was ligated, and the wound was closed. The PCB dosage was determined in preliminary experiments to identify the minimum effective concentration without causing toxicity in major organs, including the liver, lung, and kidney. Based on these results, 150 µL of PCB (6.3 mg/kg) was administered intraperitoneally seven times—once on day one before balloon injury and subsequently every two days for fourteen days. The sham group received an equivalent volume of PBS on the same schedule. The number of animals used was *n* = 5 for the sham group, *n* = 6 for the BI group, and *n* = 5 for the BI/PCB group. All quantifications were performed blinded to experimental group assignments.

### 2.5. Cell Proliferation Assay

Cells cultured in 96-well plates were exposed to specified concentrations of PDGF-BB and PCB (Sigma, St. Louis, MO, USA). After 24 h, cell proliferation was quantified using the BrdU Cell Proliferation Colorimetric ELISA kit (Promega, Madison, WI, USA), following the protocol provided by the manufacturer. PCB was dissolved in PBS at pH 7.2 before use.

### 2.6. Cell Migration Assay

Cell migration was assessed using both a three-dimensional Boyden chamber assay and a two-dimensional wound healing assay. For the Boyden chamber experiments, 5 × 10^3^ cells in 100 μL medium were seeded into the upper compartment of transwell inserts, whose lower surfaces had been pre-coated with 1% gelatin. PDGF-BB or PCB, at the indicated concentrations, was added to either the lower or upper chamber as appropriate. Following an 8 h incubation at 37 °C, cells that had migrated to the underside of the membrane were fixed, stained, and counted in five randomly selected fields per membrane at 200× magnification. For the wound healing assay, a linear scratch was generated using a cell scraper, after which cultures were treated with the designated concentrations of PDGF-BB or PCB for the specified durations. The migration distance—measured from the wound edge to the leading front of cells—was recorded, and the mean distance was calculated to represent cell motility in each dish.

### 2.7. Zymography

For gelatin zymography, equal volumes of control and treated culture media were loaded onto 10% SDS–polyacrylamide gels supplemented with 0.8% gelatin. Following electrophoresis, the gels were incubated in 2.5% Triton X-100 for 1 h to remove residual SDS, rinsed with distilled water for another hour, and subsequently placed in developing buffer (50 mM Tris–HCl, pH 7.5, 5 mM CaCl_2_, 1 μM ZnCl_2_, 0.02% sodium azide, and 1% Triton X-100) at 37 °C for 48 h. Gels were then stained with 1% Coomassie Brilliant Blue for 1 h and destained until the bands were clearly visible against the background. Images of the gels were captured, and the areas of proteolytic activity were quantified using ImageJ software (NIH, Bethesda, MD, USA).

### 2.8. Immunoblot Analysis

Carotid artery tissues were lysed in RIPA buffer with PMSF and protease inhibitors. Protein concentrations were measured using the Bradford assay (Bio-Rad, Hercules, CA, USA), and equal amounts were separated by 10% SDS-PAGE and transferred to PVDF membranes (Bio-Rad, Hercules, CA, USA). After blocking with 5% skim milk in TBS-T, membranes were incubated overnight at 4 °C with various primary antibodies. The primary antibodies were used at the following dilutions in blocking buffer: MMP2 (1:4000), MMP9 (1:1000), Talin (1:1000), and Integrin β1 (1:10,000) (Abcam, Cambridge, MA, USA), FAK (1:1000), p-FAK(Y397) (1:500), p-FAK(Y925) (1:500), Akt (1:4000), p-Akt (1:2000), ERK (1:3000), p-ERK (1:5000), P38 (1:2000), p-P38 (1:1000), PCNA (1:2000), and CDK2 (1:1000) (Cell Signaling, Beverly, MA, USA), MMP13 (Novus, Centennial, CO, USA, 1:1000), Paxillin (Millipore, Burlington, MA, USA, 1:2000), Vinculin (1:1000), cyclin D1 (1:50), and CDK4 (1:500) (Santa cruz, Dallas, TX, USA), and β-actin (Sigma, St. Louis, MO, USA, 1:10,000). Following washes, membranes were incubated with secondary antibodies, and protein bands were visualized using an enhanced chemiluminescence reagent (ECL, BIONOTE, Animal Genetics Inc., Tallahassee, FL, USA).

### 2.9. Cell Cycle Analysis

Cultured VSMCs were serum-starved in 0.1% FBS for 24 h, followed by pretreatment with PCB for 2 h. Subsequently, cells were stimulated with PDGF-BB and incubated for an additional 24 h. After treatment, cells were washed with PBS and harvested using 0.5× Trypsin-EDTA. The harvested cells were resuspended in 300 μL of PBS, and 70% ethanol was added dropwise for fixation, followed by incubation at 4 °C for 1 h with intermittent gentle vortexing. After fixation, cells were washed with PBS, treated with RNase A (10 μg/mL), and stained with propidium iodide (PI, 50 μg/mL). Cell cycle distribution was analyzed using a flow cytometer (Attune NxT, Invitrogen, Waltham, MA, USA).

### 2.10. Immunocytochemistry

Cells were seeded at 2 × 10^4^/well on cover glasses in a 24-well plate with DMEM containing 10% FBS and incubated at 37 °C for 24 h. The medium was replaced with starvation medium for 24 h, followed by pretreatment with PCB for 6 h and subsequent stimulation with 20 ng/mL PDGF-BB for 24 h. For immunofluorescence, cells were washed with cold PBS, fixed with 2% paraformaldehyde for 15 min at room temperature, washed three times, and permeabilized (or not) for 15 min with or without 0.5% Triton X-100. After blocking in PBS with 2% BSA, cells were incubated overnight at 4 °C with primary antibodies, then with secondary antibodies for 1 h at room temperature in the dark. Nuclei were counterstained with 0.2 μg/mL DAPI, and cover glasses were mounted on slides. Images were acquired using a laser scanning confocal microscope (Fluoview FV1000, Olympus, Tokyo, Japan).

### 2.11. Morphometric Analysis

Bilateral carotid arteries were harvested 14 days after surgery and fixed in 4% paraformaldehyde (PFA). After the arteries were embedded in paraffin, cross-sections (5 μM thick) were prepared from the mid-portion of the injured segment, with three to four evenly spaced sections analyzed per artery. Cross-sectional areas of the vascular layers, including the lumen (LA), intima (IA), and media (MA), were measured in at least three sections per artery (proximal, middle, distal) using ImageJ software (NIH, Bethesda, MD, USA).

### 2.12. Immunohistochemistry

Paraffin-embedded sections were deparaffinized and rehydrated through xylene and graded ethanol series prior to antigen retrieval. The sections were washed twice with 0.025% Triton X-100 in TBS for 5 min and blocked with 10% normal goat serum for 2 h at room temperature. Sections were then incubated overnight at 4 °C with primary antibodies, including MMP2 and MMP9 (Abcam, Cambridge, MA, USA; 1:50) and PCNA and Cyclin D1 (Cell Signaling, Danvers, MA, USA; 1:50). Following two washes with 0.025% Triton X-100 in TBS (5 min each), sections were incubated with the appropriate secondary antibodies for 1 h at room temperature in the dark. Nuclei were counterstained with DAPI (0.2 μg/mL, Sigma-Aldrich) for 1 min. Finally, images were captured using a laser scanning confocal microscope (Fluoview FV1000, Olympus, Tokyo, Japan). For quantification of CD1 expression, fluorescence intensity was measured separately in nuclear and cytoplasmic compartments using ImageJ software 1.54p. Nuclear CD1 values were normalized to the corresponding total CD1 intensity within the same section to account for inter-sample staining variability.

### 2.13. Statistical Analysis

All quantitative data, obtained from at least three independent experiments, were analyzed using GraphPad Prism 8.0 (La Jolla, CA, USA). Results are presented as mean ± standard deviation (SD). Comparisons between two groups were performed using Student’s *t*-test, while comparisons among multiple groups were conducted using one-way ANOVA followed by Bonferroni’s post hoc test when the ANOVA F value was significant. A two-tailed *p* < 0.05 was considered statistically significant.

## 3. Results

### 3.1. PCB Inhibits PDGF-BB-Induced VSMC Proliferation and Migration

The chemical structure of the PCB is shown in Figure 1A. Cell viability was measured by treating PCB at various concentrations for 24 h using the 3-(4,5-dimethylthiazol-2-yl)-2,5-diphenyltetrazolium bromide (MTT) assay method. After treating PCB at 0, 25, 50, 100, 150, and 200 μM for 24 h in VSMCs, cell viability and cytotoxic effects were confirmed. As shown in Figure 1B, PCB concentrations ranging from 25 to 100 μM had no significant impact on the viability of VSMCs, but at the treated concentrations of 150 and 200 μM, cytotoxicity of 38% and 69% was observed in VSMCs, respectively. Additionally, the proliferation inhibitory effect of PCB was confirmed in VSMCs induced to proliferate by PDGF-BB. Pre-treatment with non-toxic concentrations of PCB (25–100 μM) for 6 h followed by PDGF-BB treatment for another 24 h was conducted. As shown in Figure 1C, compared to the control group (white bar and black bar) treated with 0.2% DMSO or PDGF-BB, cell growth by PDGF-BB was significantly suppressed to 27%, 48%, and 55%, respectively, depending on the concentration treated with PCB. After that, the study was conducted by determining the minimum concentration of PCB treatment to 25 and the maximum concentration to 100 μM in VSMCs. Thus, as a result of confirming whether the cause of the decrease in cell growth when the PCB was treated was related to the induction of apoptosis related to caspase-3 activity, the activity of caspase-3 was reduced in the PCB treatment concentration compared to the control group that was not treated with PCB, and an increase in Bcl2, an anti-apoptosis protein, and a decrease in Bax promoting apoptosis were confirmed. Therefore, by confirming that the expression of Bcl2/Bax, which is used as an apoptosis indicator, was significantly increased at the concentration of 100 μM treated with PCB compared to the control group, it was found that the inhibitory effect of cell growth by PCB treatment was not due to cell death (Figure 1D). Furthermore, to investigate the effect of PCB in PDGF-BB-induced VSMCs migration, 2D scratch assay and 3D Boyden chamber transwell assay were performed. As shown in Figure 1E, compared to the control group, the group treated with PDGF-BB exhibited a significant increase in VSMCs migration, while a dose-dependent significant inhibition of cell migration was observed in the group treated with PCB. These results suggest that PCB inhibits PDGF-BB-induced VSMCs migration.

### 3.2. PCB Inhibits PDGF-BB Induced MMPs Expression and Activity

MMPs are involved in the migration or invasion process of various cell types, including VSMCs or cancer cells, and it has been reported that the expression of MMP2 and MMP9 in the extracellular matrix of blood vessels promotes the migration of VSMCs [6]. Therefore, to observe the changes in MMPs activities in PDGF-BB/PCB-treated VSMCs, immunofluorescence staining was performed. As seen in Figure 2A, the expression rates of MMP2 and MMP9 within VSMCs treated with PDGF-BB significantly increased compared to the control group, while they significantly decreased in cells treated with PCB. In addition, as a result of performing zymographic analysis with the culture medium of PDGF-BB/PCB-treated VSMCs, the zymolytic activities of MMP2 and MMP9 were significantly increased in the cells treated with PDGF-BB, and a concentration-dependent significant decrease was confirmed by PCB treatment (Figure 2B). However, no changes were observed in the level of MMP13 compared to the control group. Similar changes were also observed in the results of Western blot analysis (Figure 2C, see also Appendix A). These results suggest that the expression levels of MMP2 and MMP9 increased by PDGF-BB in VSMCs are inhibited by PCB, thereby inhibiting the migration of VSMCs.

### 3.3. PCB Inhibits PDGF-BB Induced VSMC Proliferation

To investigate the mechanism of the inhibitory effect of PCB proliferation in VSMCs, we confirmed cell cycle progression using fluorescence-activated cell sorting and assessed the expression levels of cell cycle-regulating proteins using immunofluorescence and Western blotting. When PCB was administered to PDGF-BB-stimulated VSMCs, a concentration-dependent increase in the G0/G1 phase cell population and a decrease in the S phase cell population were observed (Figure 3A). The protein expression levels of PCNA, CDKs, and cyclins, which are well known as markers as proteins that play essential roles in DNA synthesis and cell proliferation, were confirmed, and as shown in Figure 3B, the expression levels of PCNA, CDK2, CDK4, and Cyclin D1 increased by PDGF-BB treatment were significantly suppressed in a concentration-dependent manner by PCB treatment. Like the Western blot results, similar results were confirmed in the immunofluorescence analysis results of Figure 3C. These findings suggest that PCB inhibits the proliferation of VSMCs by inducing arrest at the G0/G1 phase and suppressing the expression of cell cycle regulatory proteins.

### 3.4. PCB Attenuates PDGF-BB-Induced the FAK-Related and p38 Signaling Pathways

There are a number of research reports that proteins such as FAK (Focal Adhesion Kinase) are involved in the proliferation and migration of VSMCs, which are known to be phosphorylated through integrin and growth factor-related signaling [30]. In this study, the level of phosphorylation in the Y397 and Y925 sites of FAK was significantly increased after treatment with PDGF-BB, which was markedly decreased by PCB treatment. In addition, by forming complexes with FAK, the protein levels of integrin β1, paxillin, vinculin, and talin related to FAK-signal transmission were also reduced in a concentration-dependent manner by PCB treatment (Figure 4A). Like the above result, a similar result was confirmed in the immunofluorescence analysis result of Figure 4B. That is, it was confirmed that the expression level of paxillin, FAK increased by PDGF-BB was significantly reduced by PCB treatment. Furthermore, while the phosphorylation levels of p-38 significantly increased after PDGF-BB treatment, they were concentration-dependently decreased by PCB treatment, whereas the phosphorylation levels of ERK, Akt, and C-Jun, which increased significantly after PDGF-BB treatment, were unaffected by PCB (Figure 4C). These results suggest that inhibition of the migration and proliferation of VSMCs by PCB treatment occurs through FAK-related and p38 pathways.

### 3.5. PCB Inhibits Neointima Formation Induced in BI Rat Carotid Arteries

Next, we evaluated the effects of PCB administration on neointimal formation in the balloon-injured (BI) model of rat carotid arteries. PCB was administered to the BI arteries of rat via intraperitoneal injection. Analysis of H&E staining from Figure 5A revealed robust neointimal formation characterized by thick concentric layers post-BI (balloon injury), compared to the control group (sham-operated animals). However, in the BI animal model group treated with PCB, a significant reduction in neointimal formation and intima/media ratio to 85–87% was observed. Subsequent immunohistochemical staining revealed an significantly increased number of MMP2-, MMP-9-, PCNA-, and Cyclin D1-positive cells in BI vessels compared to the sham group, wereas BI vessels treated with PCB showed a marked decrease in these positively satained cells (Figure 5B). These findings suggest that the protective effect of PCB on neointimal formation occurs predominantly through regulation of MMP2, 9, PCNA, and Cyclin D1 expression in VSMCs. In this study, firstly, we confirmed that PCB inhibits proliferation and migration of VSMCs through decreased expression of cell cycle regulatory proteins and MMP2, MMP9. Secondly, we found that the target signaling of PCB regulating the proliferation and migration of VSMCs occurs through FAK and p38. These results are illustrated schematically in Figure 5C.

## 4. Discussion

In this study, we demonstrated that pinocembrin (PCB), a flavonoid compound primarily found in propolis and several edible plants, significantly inhibits vascular smooth muscle cell (VSMC) proliferation and migration induced by PDGF-BB, both in vitro and in vivo. Furthermore, PCB administration markedly attenuated neointimal hyperplasia in a rat carotid artery balloon injury model. These results provide compelling evidence for the therapeutic potential of PCB in the prevention of atherosclerosis and restenosis.

Atherosclerosis is a progressive inflammatory disease characterized by lipid accumulation, endothelial dysfunction, and VSMC proliferation and migration [2,31]. VSMCs, which are normally quiescent and contractile in the healthy vascular wall, undergo phenotypic switching to a synthetic, proliferative state in response to injury or growth factors such as PDGF-BB [32]. This transition leads to increased extracellular matrix production, migration into the intima, and formation of neointimal lesions. Therefore, targeting VSMC phenotypic modulation is considered a key strategy in preventing vascular remodeling.

Our findings suggest that PCB significantly suppresses PDGF-BB-induced phosphorylation of ERK1/2 and AKT, key signaling pathways involved in VSMC proliferation and survival. Additionally, PCB inhibited MMP2 and MMP9 expression, which is critical for extracellular matrix degradation and VSMC migration. These molecular changes were accompanied by a decrease in PCNA and cyclin D1 expression, further supporting the anti-proliferative effect of PCB.

Given the adverse effects of existing synthetic cardiovascular drugs—including insomnia, hepatotoxicity, myalgia, and cognitive impairment—there is growing interest in natural compounds with safer pharmacological profiles [10,11,33]. Plant-derived polyphenolic compound, PCB, with various pharmacological activities such as antimicrobial, anti-inflammatory, antiviral, antifibrotic, antioxidant, and anticancer effects, has been suggested for its potential therapeutic applications [13]. Indeed, previous studies have highlighted the anti-inflammatory, antioxidant, and neuroprotective effects of PCB, particularly in ischemic stroke models. PCB has been shown to cross the blood–brain barrier and exert neuroprotective effects via modulation of oxidative stress, mitochondrial function, and apoptosis-related signaling pathways. Importantly, the Chinese Food and Drug Administration has approved PCB for clinical trials in ischemic stroke, and ongoing phase 2 trials suggest good tolerability and efficacy. It is also known that it protects vascular endothelial cells to prevent the progression of arteriosclerosis [22,23,24,25,26]. However, its role in cardiovascular disease, especially in modulating VSMC behavior, has remained largely unexplored.

VSMCs activate intracellular signaling pathways in response to PDGF-BB stimulation and induce cell cycle progression, cell migration, and proliferation [32]. Therefore, numerous studies are underway to investigate therapeutic strategies targeting the migration and proliferation of VSMCs mediated by PDGF-BB for the treatment and prevention of vascular diseases. Upon PDGF-BB stimulation, VSMCs activate intracellular signaling cascades such as PI3K/Akt and MAPKs (mitogen-activated protein kinases), leading to cytoskeletal remodeling and increased expression of matrix metalloproteinases (MMP2/9), which facilitate extracellular matrix degradation and cell migration [34,35,36,37]. Furthermore, it is known that the phosphorylation of FAK (Focal Adhesion Kinase) is involved in the remodeling of the cytoskeleton and has a significant effect on the migration of VSMC, and the reconstruction of F-actin to promote cell migration is regulated by the activation of FAK [8,38]. FAK is also well recognized as an important enzyme regulating the movement and proliferation of VSMCs in the pathogenesis of cardiovascular diseases [39]. Our data indicate that PCB treatment suppresses phosphorylation of FAK at Tyr397 and Tyr925, along with reduced expression of associated focal adhesion proteins, including Integrin β1, Paxillin, Talin, and Vinculin (Figure 4A). These results suggest that PCB interferes with focal adhesion dynamics, thereby reducing cytoskeletal rearrangement and motility. Additionally, the downregulation of MMP2 and MMP9 expression further supports the inhibitory role of PCB on VSMC migration, a critical process in vascular remodeling.

VSMC proliferation is driven by progression through the cell cycle, particularly DNA synthesis in the S phase [40]. In our study, PCB treatment increased the proportion of VSMCs in the G0/G1 phase and reduced S phase entry, indicating cell cycle arrest (Figure 3A). This was accompanied by downregulation of proliferation markers including PCNA, Cyclin D1, CDK2, and CDK4, all of which are upregulated by PDGF-BB stimulation [41,42]. The suppression of p38 MAPK phosphorylation by PCB appears to contribute to this effect (Figure 3B), suggesting that PCB inhibits VSMC proliferation via blockade of MAPK-mediated cell cycle progression. These results were corroborated in vivo, where PCB treatment led to significant reductions in neointimal thickness and expression of PCNA, Cyclin D1, and MMPs in the rat carotid artery model (Figure 5A–C). A limitation of the present study is that, while PCB was shown to induce cell cycle arrest and inhibit proliferation of VSMCs, the potential induction of cellular senescence was not investigated. Cellular senescence is typically characterized by increased expression of markers such as p53, p21, and p16, and can contribute to vascular aging, inflammation, and dysfunction. Given that prolonged cell cycle arrest may lead to senescence, future studies should assess these markers to distinguish between transient arrest and senescence induced by PCB treatment. Understanding this distinction is critical for evaluating the safety and long-term effects of PCB as a therapeutic agent in cardiovascular disease. Inhibiting VSMC proliferation and migration can help prevent restenosis, but it may also delay reendothelialization, which is essential for healing the blood vessel lining. Delayed reendothelialization could increase risks of thrombosis and vascular complications. Therefore, it is important to evaluate whether PCB affects endothelial cell recovery to balance preventing neointimal growth and promoting vascular repair. Future studies should explore this aspect to ensure the safety of PCB as a therapeutic agent.

PCB is known for its broad biological activities, including anti-inflammatory, antifibrotic, and antioxidant effects [13]. In myocardial infarction models, PCB improves cardiac function, reduces infarct size and arrhythmias, and protects vascular endothelial integrity [22,23,24,25,26]. Furthermore, pharmacokinetic studies report that PCB has favorable absorption and bioavailability, with suitable half-life and metabolic stability for therapeutic use [20]. Pharmacodynamically, PCB exerts multi-target effects through modulation of inflammatory signaling, apoptosis, and oxidative stress pathways [14,21]. Notably, its anti-proliferative and anti-migratory actions have been observed in cancer cells. For instance, in retinoblastoma and breast cancer cells, PCB downregulates MMP2/9 via inhibition of FAK and p38, induces cell cycle arrest, and inhibits PI3K/AKT signaling [27,28]. These known mechanisms align with our observations in VSMCs, reinforcing the potential of PCB as a vascular protective agent.

In our in vivo experiments, the selected PCB dose of 6.3 mg/kg in rats was determined through preliminary dose-finding studies as the minimal effective dose that inhibited neointimal formation without detectable hepatic or renal histopathology. Based on standard body surface area (BSA) conversion (Km_rat = 6; Km_human = 37), this corresponds to an approximate human equivalent dose (HED) of 1.02 mg/kg, which equates to about 61 mg/day for a 60 kg adult. This extrapolated dose should be interpreted cautiously, as it derives from efficacy data rather than NOAEL-based toxicology [43]. Translation to clinical application will require further pharmacokinetic and pharmacodynamic studies to evaluate absorption, distribution, metabolism, and excretion profiles, as well as detailed safety assessments including GLP toxicology, cardiovascular safety pharmacology, and potential drug–drug interactions. Considering the physiology and regulation of the cardiovascular system, the observed inhibition of VSMC proliferation and migration by PCB suggests a potential role in mitigating pathological vascular remodeling, which is a central mechanism in atherosclerosis, restenosis, and other occlusive vascular diseases. Furthermore, the known antioxidant, anti-inflammatory, and endothelial-protective properties of flavonoids, combined with the relatively favorable pharmacokinetic profile of PCB reported in the literature, support its potential as a therapeutic candidate. However, future clinical trials will be essential to confirm its efficacy and safety in humans.

It is worth noting that only male Sprague–Dawley rats were used in the in vivo experiments to minimize variability associated with hormonal fluctuations in females. While this approach improved experimental consistency, it limits the extrapolation of our findings to both sexes. Considering the established sex-specific differences in cardiovascular physiology and pathology, future studies including both male and female animals will be essential to evaluate the therapeutic potential of PCB in a broader patient population.

Regarding target specificity, our data indicate that PCB primarily exerts its inhibitory effects on PDGF-BB–induced signaling cascades downstream of receptor activation, particularly FAK and p38 MAPK. While our study did not directly assess other growth factor stimuli, the molecular pathways modulated by PCB—such as cytoskeletal reorganization and MMP regulation—are common to multiple receptor tyrosine kinase (RTK)-mediated responses. Therefore, it is plausible that PCB’s inhibitory actions could extend to other proliferative and migratory stimuli, such as basic fibroblast growth factor (bFGF), epidermal growth factor (EGF), or vascular endothelial growth factor (VEGF). Moreover, given the structural similarities among flavonoids, compounds with related backbones—such as pinostrobin or galangin—may exhibit comparable bioactivity. Nevertheless, specificity profiling against various ligands and structurally related molecules will be essential to define the selectivity and potential therapeutic niche of PCB.

While our findings support PCB’s role in modulating key pathways involved in VSMC proliferation and migration, several limitations remain. First, further studies using genetically modified atherosclerotic models (e.g., apoE-/- mice) are needed to evaluate long-term effects and safety. Second, although we identified major signaling targets such as FAK and p38 MAPK, a comprehensive proteomic or transcriptomic analysis could reveal additional targets. Lastly, pharmacokinetic and toxicological profiling in disease models will be necessary to establish translational feasibility. Nevertheless, our study provides new insight into the vascular protective effects of PCB and supports its development as a promising natural therapeutic for atherosclerosis and restenosis.

## 5. Conclusions

PCB is one of the major flavonoids abundantly present in propolis and can be obtained in pure form with high stability. Owing to its favorable safety profile, relatively low production cost, and promising pharmacological activities, PCB represents an attractive candidate for future therapeutic applications. The findings of this study demonstrate that PCB effectively inhibits abnormal proliferation and migration of vascular smooth muscle cells, indicating its potential in the prevention, treatment, and amelioration of vascular pathologies such as atherosclerosis, vascular stenosis, and other secondary cardiovascular disorders. Given its multi-targeted actions and potential for clinical translation, PCB could serve as a valuable therapeutic agent either as a monotherapy or in combination with existing cardiovascular drugs. Nevertheless, comprehensive preclinical investigations and well-designed clinical trials are warranted to further elucidate its pharmacokinetic profile, long-term safety, and efficacy in human cardiovascular disease prevention and management.

## Figures and Tables

**Figure 1 biomolecules-15-01325-f001:**
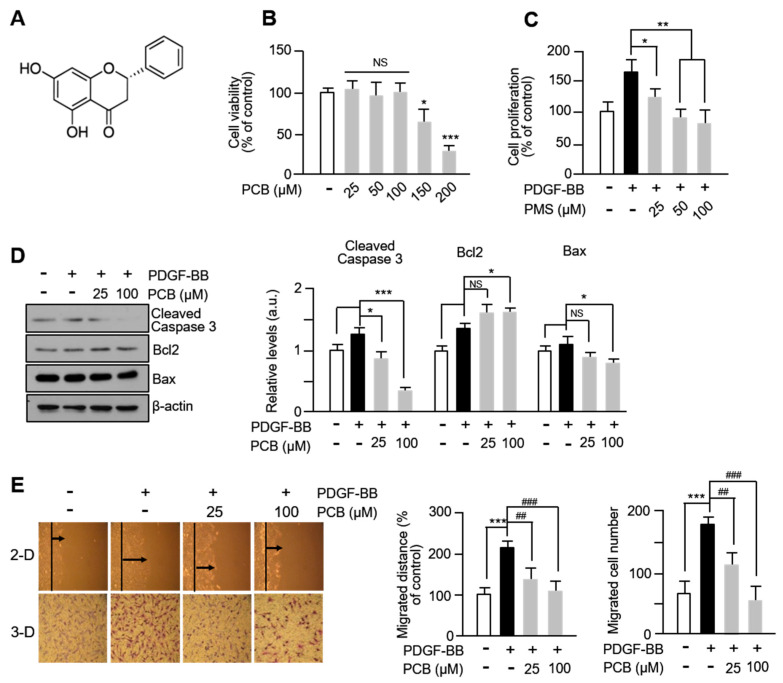
PCB inhibits PDGF-BB-induced VSMC proliferation and migration. (**A**) Chemical structure of PCB. (**B**) Cytotoxicity of PCB was evaluated using the MTT assay. Serum-starved VSMCs were treated with PCB (25~200 μM) for 24 h. (**C**) The effects of PCB on VSMC proliferation were evaluated using the BrdU incorporation assay. Serum-starved VSMCs were treated with PCB (25~100 μM) for 6 h and then incubated with PDGF-BB (25 ng/mL) for 24 h. (**D**) The effects of PCB on Bcl-2/Bax proteins in VSMCs. The Bcl-2/Bax ratio was calculated from the Bax and Bcl-2 over β-actin ratios. (**E**) Effects of PCB on VSMC migration were measured using a wound healing assay and 3D Boyden chamber assay. VSMCs were wounded (black lines) and treated with PDGF-BB or PCB. After 24 h, the migrated cells were measured using the ImageJ Software 1.54p. The black arrow represents the distance of the most migrated cells. The bottom shows the number of migrated cells. Representative images are shown from at least five independent experiments, taken at the time of 8 h after seeding. Mean ± SD (*n* = 5). * *p* < 0.05, ** *p* < 0.01, and *** *p* < 0.001 PCB versus PDGF-BB alone (**B**–**D**); *** *p* < 0.001 PDGF-BB versus control and ## *p* < 0.01, ### *p* < 0.001 PCB versus PDGF-BB alone (**E**); NS, no significance.

**Figure 2 biomolecules-15-01325-f002:**
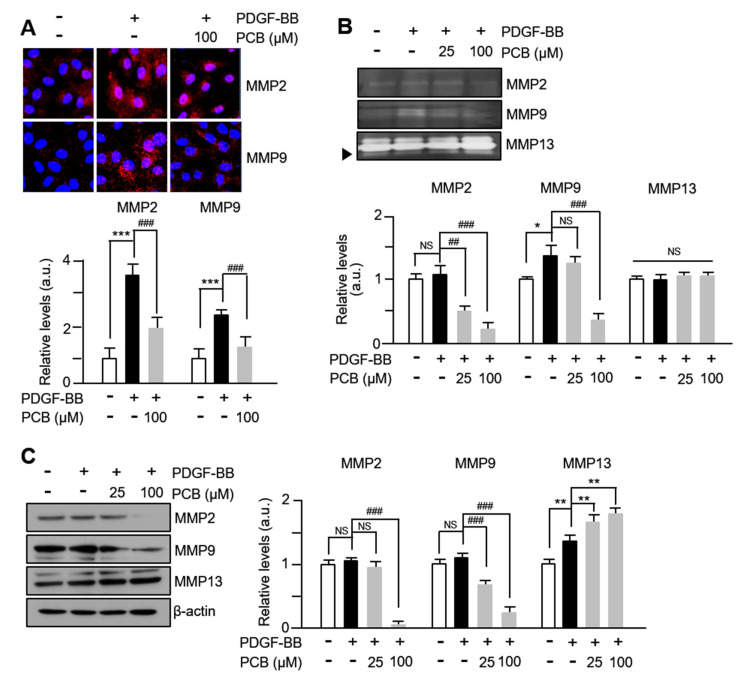
PCB inhibits PDGF-BB induced MMPs expression and activity. (**A**) The expression of MMPs proteins (MMP2, 9, red) were measured by immunocytochemical staining. Nuclei were stained with DAPI (blue). Representative images are shown from at least three independent experiments. (**B**) The MMPs activities were evaluated using gelatin zymography. Serum-starved VSMCs were incubated with PCB (25~100 μM) for 2 h, followed by PDGF-BB (25 ng/mL) treatment for 24 h. The media was collected and used for these assays. (**C**) Effects of PCB on the expression of MMPs. Serum-starved VSMCs were incubated with PCB (25~100 μM) for 2 h, followed by PDGF-BB (25 ng/mL) treatment for 24 h. The band densities were normalized to β-actin signals. Representative images are shown from at least three independent experiments. Mean ± SD (*n* = 4). * *p* < 0.05, ** *p* < 0.01, and *** *p* < 0.001 PDGF-BB versus control; ## *p* < 0.01 and ### *p* < 0.001 PCB versus PDGF-BB alone; NS, no significance.

**Figure 3 biomolecules-15-01325-f003:**
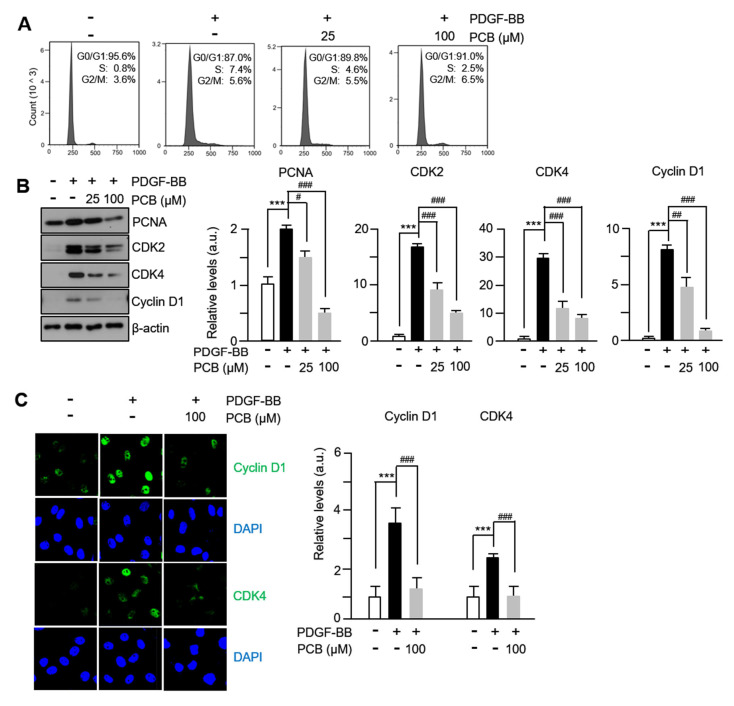
PCB inhibits PDGF-BB induced proliferation. (**A**) Effects of PCB on cell cycle progression. Serum-starved VSMCs were incubated with PCB (25~100 μM) for 2 h followed by 25 ng/mL of PDGF-BB treatment for 24 h. After DNA staining with PI, the cells were analyzed by flow cytometry. Each value shown was derived from counting >10,000 events and the numbers of cells (G0/G1, S, and G2/M phases) are expressed as % of 10,000 events (*n* = 5). (**B**) Inhibitory effect by PCB of PDGF-BB-stimulated expression of cell cycle regulatory proteins. Serum-starved VSMCs were incubated with PCB (25~100 μM) for 2 h, followed by PDGF-BB (25 ng/mL) treatment for 24 h. The band densities were normalized to β-actin signals. (**C**) The expression of cell cycle regulatory proteins (Cyclin D1, CDK4, green) were measured by confocal microscopy. Nuclei were stained with DAPI (blue). Representative images are shown from at least three independent experiments. Mean ± SD (*n* = 4). *** *p* < 0.001 PDGF-BB versus control; # *p* < 0.05, ## *p* < 0.01, and ### *p* < 0.001 PCB versus PDGF-BB alone.

**Figure 4 biomolecules-15-01325-f004:**
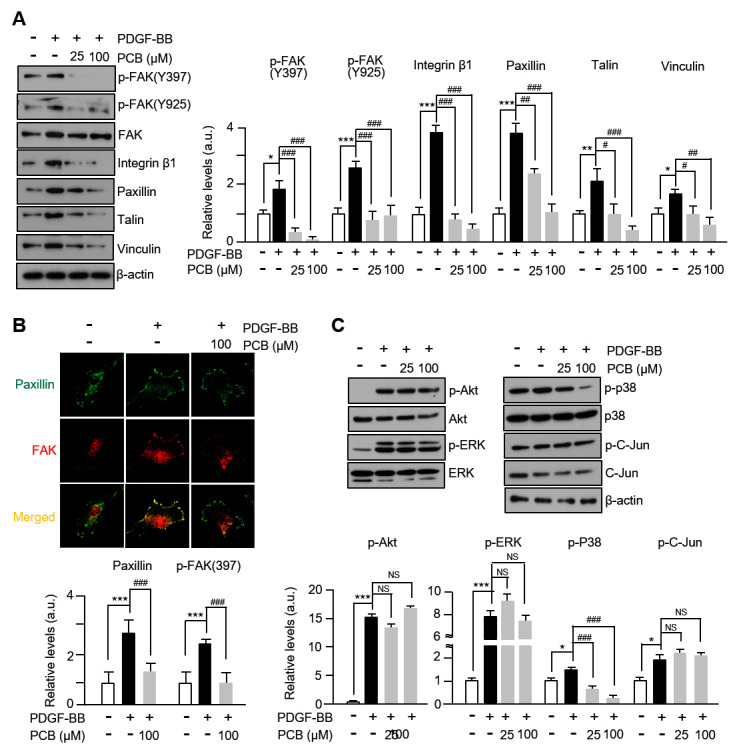
PCB attenuates PDGF-BB-induced the FAK-related and p38 signaling pathways. (**A**) Effects of PCB on the expression of FAK-related proteins in VSMCs. Serum-starved VSMCs were incubated with PCB (25~100 μM) for 2 h followed by 25 ng/mL of PDGF-BB treatment for 24 h. Protein levels were normalized to β-actin signals with the exception of p-FAK (Y397, Y925), which were normalized to their total FAK levels. (**B**) Extracellular levels of FAK-related proteins (Paxillin, green; FAK, red) were measured by confocal microscopy. (**C**) Effects of PCB on the expression of MAPK. Serum-starved VSMCs were incubated with PCB (25~100 μM) for 2 h followed by 25 ng/mL of PDGF-BB treatment for 30 min. The band densities phosphorylated proteins were normalized to those of total protein expression. Representative images are shown from at least three independent experiments. Mean ± SD (*n* = 4). * *p* < 0.05, ** *p* < 0.01, and *** *p* < 0.001 PDGF-BB versus control; # *p* < 0.05, ## *p* < 0.01, and ### *p* < 0.001 PCB versus PDGF-BB alone; NS, no significance.

**Figure 5 biomolecules-15-01325-f005:**
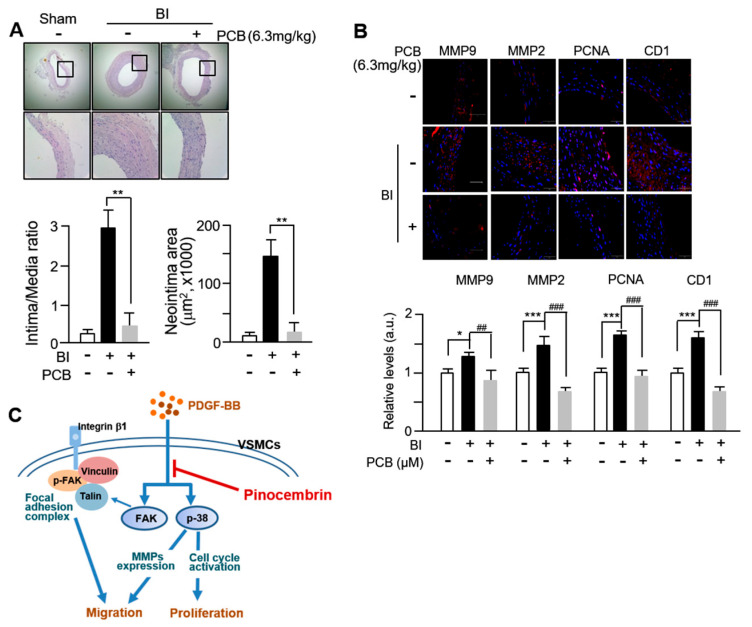
PCB inhibits neointima formation induced in BI rat carotid arteries. (**A**) Representative hematoxylin and eosin (HE)-stained sections of carotid arteries from sham, balloon injury (BI), and BI with PCB treatment (BI/PCB) groups. Sections were taken from the mid-portion of the injured segment to ensure anatomical consistency. Morphometric analysis was performed on 3–4 evenly spaced cross-sections per artery, with *n* = 5, 6, and 5 animals in the sham, BI, and BI/PCB groups, respectively. Neointimal area and intima/media (I/M) ratio were quantified as described in the Methods. ** *p* < 0.01 PCB versus BI alone. (**B**) Immunofluorescence (IF) staining of CD1, MMP2, MMP9, and PCNA was performed on sections immediately adjacent to the HE-stained regions, to maintain comparable anatomical locations. Nuclei were counterstained with DAPI (blue). CD1 expression was evaluated in both cytoplasmic and nuclear compartments. Quantitative IF analysis was normalized to the corresponding total protein levels in each sample, which eliminates potential variability from loading or staining differences; thus, no additional adjustment was necessary. Data are presented as mean ± SD (*n* = 4). * *p* < 0.05 and *** *p* < 0.001 BI alone versus Sham; ** *p* < 0.01, ## *p* < 0.01 ### *p* < 0.001 PCB vs. BI alone. (**C**) Schematic diagram showing the proposed inhibitory mechanism of PCB on PDGF-BB-induced proliferation and migration of vascular smooth muscle cells (VSMCs).

## Data Availability

The original contributions presented in this study are included in the article. Further inquiries can be directed to the corresponding author.

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
