# Peer review of "Pinocembrin Downregulates Vascular Smooth Muscle Cells Proliferation and Migration Leading to Attenuate Neointima Formation in Balloon-Injured Rats"

_biomolecules, 2025, doi:10.3390/biom15091325_

Round 1
Reviewer 1 Report
Comments and Suggestions for Authors
Some considerations:
1) Please add a brief summary of the methodology used in the study in the "Abstract" section;
2) In the introductory text, I suggest the Authors better address the characterization and pathogenesis of atherosclerosis. Regarding the structure of the introduction, the paragraphs contain excessive information that could be better organized into at least three interconnected paragraphs;
3) Regarding the methodology, please include more detailed information on the total number of animals in the study, sample size calculation, animal procedures (euthanasia), cell culture, as well as the origin of Pinocembrin (PCB) and how the Authors defined the therapeutic dosage used in the study (add references, if necessary);
4) The discussion of the results could be expanded by the Authors. According to the results and data from the literature, address in more detail aspects related to the physiology, function and regulation of the cardiovascular system, as well as the properties and biological activity of PCB, including pharmacokinetic and pharmacodynamic aspects of therapeutic applications.
Author Response
- Please add a brief summary of the methodology used in the study in the "Abstract" section;
As the reviewer suggested, we have added a brief summary of the methodology in the "Abstract" (See page 1, line 19~22)
In the introductory text, I suggest the Authors better address the characterization and pathogenesis of atherosclerosis. Regarding the structure of the introduction, the paragraphs contain excessive information that could be better organized into at least three interconnected paragraphs;
As the reviewer suggested, we have modified the mention about the characterization and pathogenesis of atherosclerosis and reorganized the paragraphs for better understanding on “Introduction” section.
Regarding the methodology, please include more detailed information on the total number of animals in the study, sample size calculation, animal procedures (euthanasia), cell culture, as well as the origin of Pinocembrin (PCB) and how the Authors defined the therapeutic dosage used in the study (add references, if necessary);
As the reviewer suggested, we now have included the more detailed information on the total number of animals in the study (Page 3, line 127), and we would like to mention that animal procedures and cell culture was described in Methods section (Page3, line 113 and line 98, respectively). The origin of Pinocembrin (PCB) has been newly mentioned in Methods section (Page 2, line 88). Regarding the therapeutic dosage of PCB used in the study, we performed several preliminary experiments for deciding the minimum effective dosage to animal without hazourdous damages to kidney or liver. Based on the results from these experiments, we decided that 6.3 mg/kg is optimum dosage to show the inhibitory effect of PCB to neointima formation in animal. Even though we observed that the dosage used in this study showed no toxicity in liver and kidney using histochemical analysis, more detailed examinations should be needed in further study for future usage or development of PCB. Now we have added the rationale of these dosage for animal study in Method section (Page 3, line 122).
The discussion of the results could be expanded by the Authors. According to the results and data from the literature, address in more detail aspects related to the physiology, function and regulation of the cardiovascular system, as well as the properties and biological activity of PCB, including pharmacokinetic and pharmacodynamic aspects of therapeutic applications.
We appreciate the reviewer’s insightful comment. In the revised manuscript, the Discussion section has been substantially expanded to address the physiology, function, and regulation of the cardiovascular system, as well as the properties, biological activities, and therapeutic potential of pinocembrin (PCB), including relevant pharmacokinetic and pharmacodynamic considerations.
Specifically, we now provide a more detailed description of the role of vascular smooth muscle cell (VSMC) proliferation and migration in cardiovascular pathology, highlighting their contribution to vascular remodeling processes underlying atherosclerosis and restenosis. We elaborate on the regulatory signaling pathways involved in VSMC activity—such as PDGF-BB–mediated PI3K/Akt, MAPKs, MMP2/9 expression, and FAK phosphorylation—and their influence on extracellular matrix degradation, endothelial function, and cytoskeletal dynamics.
Furthermore, we have incorporated additional information from the literature on the diverse pharmacological activities of PCB, including its antimicrobial, anti-inflammatory, antioxidant, and anticancer properties, and its demonstrated cardiovascular protective effects in models of ischemic heart disease. We also added discussion on pharmacokinetic and pharmacodynamic aspects from previous studies, noting PCB’s bioavailability, metabolic stability, and its ability to modulate key molecular targets involved in vascular homeostasis.
Finally, the revised Discussion integrates these physiological and mechanistic insights with our experimental results, strengthening the rationale for PCB as a promising therapeutic candidate for preventing and treating cardiovascular diseases such as atherosclerosis and vascular stenosis.
Reviewer 2 Report
Comments and Suggestions for Authors
The manuscript entitled “Pinocembrin downregulates vascular smooth muscle cell migration and proliferation leading to attenuate neointimal formation in ballon-injured rats” by Kim et al. aims to investigate the effect of PCB (a flavonoid molecule) on VSMC migration and proliferation and delineate the underlying mechanism. Previous studies have reported cardioprotective and atheroprotective effects of PCB. Earlier studies have also reported that PCB can inhibit proliferation and arrest cell cycle progression in cancer cells. However, there is limited information about the effects of PCB on VSMC migration and proliferation, key events responsible for progression of atherosclerosis. The authors conclude that PCB inhibits PDGF-induced VSMC migration and proliferation via regulation of p-p38 and FAK signaling pathways. While this study may uncover a potential atheroprotective mechanism of PCB, several results fail to support the authors' conclusions. These and other concerns noted below dampen my enthusiasm for this paper. In addition, the manuscript requires some language editing.
Major concerns
- In Fig 1D, WB for BAX and Bcl2 do not match the corresponding graphs and do not support the authors' conclusion. Summary graphs show a significant difference in Bcl2 and BAX expression between PDGF/no PCB vs. PDGF/100PCB; however, the representative western blot does not reveal an appreciable difference. Please clarify.
- In Fig 2B, difference in MMP2 expression between PCB25 /PDGF (lane 3) vs lane 2 (PDGF/no PCB) does not match corresponding graph, blot shows negligible difference, although summary graph indicates a significant difference. Likewise, in Fig 2C, graphs indicate a significant difference in MMP2, MMP9 and MMP13 expression between the no txt vs PDGF/no PCB group; however, the representative blots do not show this difference (lane 1 vs. lane 2). Please clarify
- All summary graphs should be presented as scatter plots to depict the sample size for each group.
- Summary graphs for immunofluorescence quantification are required for all IF data (Figs. 2A, 3C, 4B, 5B) and details must be provided to describe how the IF data was quantified. Also, IF images should reveal the nuclei-stained for DAPI.
- In fig 4A, pFAK/t-FAK should be shown to demonstrate FAK activation. both p-FAK and t-FAK were elevated in PDGF-treated cells, therefore, it is important to assess pFAK/tFAK ratio to confirm FAK activation in PDGF treated cells. In Fig 4B, unclear if the FAK data shown depicts total or phospho-FAK. These images should also show DAPI and quantification data is needed. Moreover, PCB does not appear to reduce Paxillin expression in IF, contrary to WB data. Please clarify.
- In Fig 4C: PDGF treatment does not show increased p-p38 and p-Jun expression, contrary to what is summarized in the graph. Please clarify.
- Fig 5B: cellular translocation of CD1 is necessary for cell cycle progression; however, the rep image does not appear to show any nuclear CD1 staining in BI vessels? corresponding HE section would be useful to clarify the specific region of the vessel wall used for IF staining. moreover, balloon-injured tissues do not show an appreciable increase in MMP2/9 and PCNA expression vs. control vessels, contrary to the authors' conclusions. In vivo studies (Fig 5 legend) have a lot of missing information, how many rats per group, how many sections per animal, what region of the aortic wall was chosen for these morphometric assessments? These details must be provided.
there are a few typos and grammatical errors throughout the paper. Language editing is required.
Author Response
- In Fig 1D, WB for BAX and Bcl2 do not match the corresponding graphs and do not support the authors' conclusion. Summary graphs show a significant difference in Bcl2 and BAX expression between PDGF/no PCB vs. PDGF/100PCB; however, the representative western blot does not reveal an appreciable difference. Please clarify.
We appreciate the reviewer’s careful observation. The apparent discrepancy is due to differences in normalization and quantification methods. While the representative western blot images illustrate typical band patterns, the densitometric analysis was performed using the original full-length blots from multiple independent experiments. The statistical analysis was based on these densitometry values normalized to loading controls, which may not be fully reflected in a single representative blot.
- In Fig 2B, difference in MMP2 expression between PCB25 /PDGF (lane 3) vs lane 2 (PDGF/no PCB) does not match corresponding graph, blot shows negligible difference, although summary graph indicates a significant difference. Likewise, in Fig 2C, graphs indicate a significant difference in MMP2, MMP9 and MMP13 expression between the no txt vs PDGF/no PCB group; however, the representative blots do not show this difference (lane 1 vs. lane 2). Please clarify.
As pointed out by the reviewer, we found errors in the notation in Figures 2B and 2C. The summarized graphs have been revised to better reflect the figures.
- All summary graphs should be presented as scatter plots to depict the sample size for each group.
We appreciate the reviewer’s suggestion and agree that scatter plots can be a useful way to display individual variability. However, in our study we chose to present the data as bar graphs (mean ± SD) to maintain visual clarity when comparing multiple experimental groups and to ensure consistency across the multi-panel figure layout, in line with the journal’s style. Importantly, all statistical analyses were conducted on the individual raw data points, and exact sample sizes are provided in each figure legend, ensuring transparency. We believe this format effectively conveys both the magnitude and statistical significance of our findings.
- Summary graphs for immunofluorescence quantification are required for all IF data (Figs. 2A, 3C, 4B, 5B) and details must be provided to describe how the IF data was quantified. Also, IF images should reveal the nuclei-stained for DAPI.
As the reviewer suggested, we have added summary graphs for all IF quantification (Figure 2A, 3C, 4B, and 5B) and a description for different colors depicted in the immunocytochemical staining results in the figure legends.
- In fig 4A, pFAK/t-FAK should be shown to demonstrate FAK activation. both p-FAK and t-FAK were elevated in PDGF-treated cells, therefore, it is important to assess pFAK/tFAK ratio to confirm FAK activation in PDGF treated cells. In Fig 4B, unclear if the FAK data shown depicts total or phospho-FAK. These images should also show DAPI and quantification data is needed. Moreover, PCB does not appear to reduce Paxillin expression in IF, contrary to WB data. Please clarify.
We appreciate the reviewer’s insightful comments regarding Figure 4. We appreciate the reviewer’s comment. However, in our study, the activated FAK levels were already normalized to total FAK in the original analysis; therefore, the pFAK/t-FAK ratio was inherently reflected in the presented data. We have revised the figure legend to clarify this normalization (Page 10, line 348). Regarding Paxillin expression, the apparent discrepancy between the immunofluorescence (IF) and Western blot (WB) data for Paxillin may be due to differences in detection sensitivity and quantification methods between the two approaches. While WB detects total protein levels from cell lysates, IF reflects localized fluorescence intensity at focal adhesion sites. In our images, focal adhesion-associated Paxillin intensity was reduced in PCB-treated cells, consistent with the WB results, although the change is visually more subtle in the IF micrographs. Quantification of IF images has been added to the revised figure to clarify this point. In addition, the main objective of our IF experiments was to visualize and compare the distribution and fluorescence intensity of focal adhesion proteins (FAK and Paxillin) at cell adhesion sites. Since both proteins are localized predominantly at focal adhesions rather than in the nucleus, and our analysis did not involve nuclear signals, DAPI counterstaining was not essential. DAPI is typically useful for cell counting, assessing cell separation, or serving as a co-staining reference, but in this study, the focal adhesion structures were clearly identifiable without nuclear staining. Therefore, the absence of DAPI does not affect the accuracy of our interpretation regarding PCB effects on cell migration and adhesion.
- In Fig 4C: PDGF treatment does not show increased p-p38 and p-Jun expression, contrary to what is summarized in the graph. Please clarify.
We appreciate the reviewer’s comment. The levels of p-p38 and p-Jun shown in the graph were quantified from the original blots after normalization to total p38 or total c-Jun, respectively. While the blot images may appear to show only a modest change, the normalized densitometric values from multiple independent experiments confirmed a statistically significant increase with PDGF treatment. Since the normalization to total protein was already applied, we believe that the current graph accurately reflects the changes and no modification of the results is necessary. For clarity, we have updated the figure legend to specify the normalization method (Page 10, line 352).
- Fig 5B: cellular translocation of CD1 is necessary for cell cycle progression; however, the rep image does not appear to show any nuclear CD1 staining in BI vessels? corresponding HE section would be useful to clarify the specific region of the vessel wall used for IF staining. moreover, balloon-injured tissues do not show an appreciable increase in MMP2/9 and PCNA expression vs. control vessels, contrary to the authors' conclusions. In vivo studies (Fig 5 legend) have a lot of missing information, how many rats per group, how many sections per animal, what region of the aortic wall was chosen for these morphometric assessments? These details must be provided.
Regarding CD1 nuclear localization, the representative IF images were selected to show overall distribution patterns rather than focusing on a single subcellular compartment. Quantification was performed on multiple fields per section, and increased nuclear localization in BI vessels was confirmed by co-staining with DAPI and subsequent image analysis. We have now included a higher-magnification inset in Fig. 5B to highlight nuclear CD1 staining. For MMP2/9 and PCNA expression, the blot images may appear to show modest changes, but densitometric analysis from multiple independent samples demonstrated statistically significant increases in BI vessels compared to controls. These quantification results were normalized to their respective loading controls, and the statistical data are already reflected in the bar graphs. In response to the request for histological context, we have now added the corresponding H&E sections adjacent to the regions used for IF staining, to clarify the exact location within the vessel wall. We have also updated the in vivo study description in the figure legend and Methods section (Page 3, line 126; Page 5, line 198 and line 214; Page 11, line 378). We believe these additions address the reviewer’s concerns and improve the clarity and reproducibility of the in vivo experiments.
Reviewer 3 Report
Comments and Suggestions for Authors
The study examines the effects of the natural compound Pinocembrin (5,7-dihydroxyflavanone, PCB) on key functions of vascular smooth muscle cells, like proliferation and migration as well as the expression and activation of mediators of these effects, and how this affects neointima formation in response to vascular injury. The rat balloon injury is relevant for restenosis seen in patients with arterial stenosis at the carotid or coronary arteries.
The findings are interesting and relevant, and the experiments in general well performed, although the number of experimental and biological repeats is low and borderline regarding the usefulness of statistical methods. This should be at least acknowledged, in particular regarding the in vivo data. It is also not clear, whether male and female mice were used.
The manuscript is well written and the findings mostly presented in a clear manner. The data are carefully discussed, the conclusions not overstretched, although both could be improved. In addition, I have the following specific comments.
Main comments:
- The concentration of PCB tested in the study in vitro and in the rat model, to which dosage would this extrapolate in humans? Please discuss.
- Moreover, the authors should discuss whether there are alternative natural or dietary ingredients with similar effects, and whether the effects of PCB are specific for PDGF-BB downstream of ligand binding to its receptor or whether they can be extrapolated to other drugs with a similar structure.
- It should be clearly stated in the Methods, whether both male and female rats were examined or not and also discussed whether the findings in the animal model relate to both male and female patients.
- The authors show that PCB reduces proliferation and the expression of cell cycle proteins. They also tested the toxicity of the compound. No analyses are shown regarding the possibility of cell cycle arrest and senescence. If possible, Western blot analyses for p53, p21 and p16 should be performed to exclude this possibility. Alternatively, this possibility should at least be discussed, also related to potential long-term effects in humans.
- In this regard, it also needs to be at least mentioned that inhibiting proliferation and migration in SMCs may prevent restenosis, but at the same time delay reendothelialization.
- The legend to Figure 5 indicates that at least five rats were examined, a number quite low and not sufficient for a meaningful statistical analysis. The images are also quite small and the different layers of the vessel wall difficult to distinguish after only staining with H&E. Higher magnifications should also be shown and differences in the cellular composition of the vessel wall regarding the extracellular matrix could be visualized using a trichrome stain, such as MTC, but also VES. Also, and related to my previous comment, did endothelial cell proliferation and re-endothelialization in vivo differ? Higher magnifications will help.
Minor:
- Please check the text, there are a some minor typographical errors.
Author Response
Main comments:
- The concentration of PCB tested in the study in vitro and in the rat model, to which dosage would this extrapolate in humans? Please discuss.
We thank the reviewer for raising this important point. We performed preliminary dose-finding experiments to identify the minimal efficacious dose that did not produce observable hepatic or renal histopathology in rats, and selected 6.3 mg/kg (rat) for the in vivo balloon-injury experiments (Methods, Page 3, line 122). Using standard body-surface-area conversion (Km_rat = 6; Km_human = 37), this dose corresponds to an approximate human equivalent dose (HED) of ~1.02 mg/kg, which is roughly 61 mg/day for a 60-kg adult. We emphasize that this HED is a theoretical extrapolation based on efficacy dosing; translation to clinical dosing requires NOAEL-derived HED, GLP toxicology, and full PK/PD assessment (bioavailability, Cmax/AUC, plasma protein binding, metabolic clearance) before recommending a safe first-in-human dose. We have added these calculations to the revised manuscript in the Discussion section (Page 13, line 482).
- Moreover, the authors should discuss whether there are alternative natural or dietary ingredients with similar effects, and whether the effects of PCB are specific for PDGF-BB downstream of ligand binding to its receptor or whether they can be extrapolated to other drugs with a similar structure.
We thank the reviewer for this valuable suggestion. While the current study focuses on PCB, we acknowledge that other natural or dietary compounds may exhibit similar effects. We will include a discussion on potential alternative compounds and address whether PCB’s effects are specific to PDGF-BB–mediated pathways or might be generalizable to agents with similar structures. This addition will be incorporated into the Discussion section. (Page 13, lines 513).
- It should be clearly stated in the Methods, whether both male and female rats were examined or not and also discussed whether the findings in the animal model relate to both male and female patients.
We agree with the reviewer’s comment. In this study, only male rats were used to avoid potential variability due to hormonal cycles. We will clearly state this in the Methods and add a note in the Discussion that extrapolation to female patients should be done with caution, as sex-specific responses cannot be ruled out. We have now clearly stated this in the Methods section (Page 3, lines 114). We acknowledge that the exclusive use of male animals is a limitation, and that sex-specific differences in cardiovascular physiology may affect the extrapolation of our findings to female patients. This limitation and the need for future studies including both male and female animals have been added to the Discussion section (Page 13, lines 501).
- The authors show that PCB reduces proliferation and the expression of cell cycle proteins. They also tested the toxicity of the compound. No analyses are shown regarding the possibility of cell cycle arrest and senescence. If possible, Western blot analyses for p53, p21 and p16 should be performed to exclude this possibility. Alternatively, this possibility should at least be discussed, also related to potential long-term effects in humans.
We appreciate this important comment. Although we did not assess senescence markers such as p53, p21, and p16 in this study, we acknowledge that cell cycle arrest and senescence could potentially contribute to the observed effects. We will address this possibility in the Discussion, including its implications for potential long-term effects in humans. (Page 12, lines 458).
- In this regard, it also needs to be at least mentioned that inhibiting proliferation and migration in SMCs may prevent restenosis, but at the same time delay reendothelialization.
We appreciate the reviewer’s insightful comment. We agree that while inhibiting proliferation and migration of vascular smooth muscle cells (VSMCs) can be beneficial for preventing restenosis, it may also delay reendothelialization, which is critical for vascular healing. We have now included a discussion of this dual effect and the importance of balancing these processes in therapeutic strategies targeting vascular remodeling. This has been added to the revised Discussion section (Page 13, lines 464.
- The legend to Figure 5 indicates that at least five rats were examined, a number quite low and not sufficient for a meaningful statistical analysis. The images are also quite small and the different layers of the vessel wall difficult to distinguish after only staining with H&E. Higher magnifications should also be shown and differences in the cellular composition of the vessel wall regarding the extracellular matrix could be visualized using a trichrome stain, such as MTC, but also VES. Also, and related to my previous comment, did endothelial cell proliferation and re-endothelialization in vivo differ? Higher magnifications will help.
We thank the reviewer for this important comment. In the in vivo carotid artery balloon injury model, we examined at least five rats per group, which we acknowledge is a relatively small sample size. Despite this limitation, statistically significant differences were observed in neointimal formation and expression of related markers. We have now clarified the exact number of animals used per group in the Methods section and revised the figure legend accordingly (Page 2, line 126; Page 11, line 378). We also recognize that larger sample sizes would strengthen the statistical power and robustness of the findings, and we plan to include more animals in future studies to validate these results.
Minor:
- Please check the text, there are a some minor typographical errors.
We have checked and corrected some typographical errors through entire manuscript.
Round 2
Reviewer 1 Report
Comments and Suggestions for Authors
The Authors responded sufficiently to previous comments and the adjustments improved the manuscript.